# Metathesis Cyclopolymerization Triggered Self-Assembly of Azobenzene-Containing Nanostructure

**DOI:** 10.3390/molecules25173767

**Published:** 2020-08-19

**Authors:** Wei Song, Jiamin Shen, Xiang Li, Jinhui Huang, Liang Ding, Jianhua Wu

**Affiliations:** 1Department of Polymer and Composite Material, School of Materials Engineering, Yancheng Institute of Technology, Yancheng 224051, China; 18652367440@163.com (J.S.); bhf12345@126.com (X.L.); h18861901716@163.com (J.H.); 2Department of Materials, College of Physics, Mechanical and Electrical Engineering, Jishou University, Jishou 416000, China

**Keywords:** metathesis cyclopolymerization, double-stranded polymer, poly(1,6-heptadiyne), self-assembly, azobenzenes

## Abstract

Azobenzene (AB) units were successfully introduced into poly(1,6-heptadiyne)s in order to ensure smooth synthesis of double- and single-stranded poly(1,6-heptadiyne)s (**P1** and **P2**) and simultaneously realize the self-assembly by Grubbs-III catalyst-mediated metathesis cyclopolymerization (CP) of AB-functionalized bis(1,6-heptadiyne) and 1,6-heptadiyne monomers (**M1** and **M2**). Monomers and polymers were characterized by ^1^H NMR, mass spectroscopy, and GPC techniques. The double-stranded poly(1,6-heptadiyne)s exhibited a large scale of ordered ladder nanostructure. This result was attributed to the π−π attractions between end groups along the longitudinal axis of the polymers and van der Waals interactions between the neighboring polymeric backbones. While the Azo chromophore connected in the side chain of **P2** induced conformation of micelles nanostructure during the CP process without any post-treatment. Furthermore, the photoisomerization of Azo units had an obviously different regulatory effect on the conjugated degree of the polymer backbone, especially for the single-stranded **P2**, which was attributed to the structural differences and the interaction between AB chromophores in the polymers.

## 1. Introduction

In the past ten years, the fabrication of polymeric nanoscale morphologies by polymerization-induced self-assembly (PISA) method has attracted considerable attention [1,2,3,4,5] because the polymers are synthesized, and the self-assembly of the resulted polymers is completed simultaneously. PISA methods predominantly use radical polymerization processes, such as reversible addition-fragmentation termination (RAFT) polymerization [6,7,8,9]. Metathesis of cyclopolymerization (CP) of terminal 1,6-heptadiynes is one of the most efficient and powerful tools for the synthesis of conjugated polyacetylene. Extensive studies by our group and Choi and Xie have employed CP for the formation of a series of nanostructures in solution, ranging from spheres to vesicles, nanotubes, nanofibers, and ladderphane nanostructures [10,11,12,13,14,15,16].

Azobenzenes (AB)-containing materials have been widely utilized as AB can be reversibly photoisomerized between the trans and cis forms by ultraviolet-visible and light irradiation, which makes various novel optical applications and aggregation morphology adjusting [17,18,19,20,21,22,23,24]. Lots of reports have focused on the incorporation of azobenzene chromophores into polymer main chains and side chains to induce diverse topologic structures as functional substituent groups [25,26,27,28]. Our group has reported the morphology and optical control of AB-containing polymers by reversible *trans*-*cis* isomerization phenomenon [29,30,31]. Nevertheless, introducing AB units into conjugated polyacetylenes (PAs) and investigating the effect of the AB chromophores on the morphological and photophysical properties of PAs are still rare so far.

Polymeric ladderphane is a step-like structure comprising multiple layers of linkers covalently connected to two or more polymeric backbones [32]. This long-range of ordered morphological patterns result from interactions between phenyl head and ethyl vinyl ether tail end groups via π-π stacking along the longitudinal axis of the polymer and van der Waals interaction between neighboring backbones of polymers [32]. Consequently, ladderphane polymers have greater resistance to irradiation, as well as thermal and chemical degradation, in comparison to their single-stranded counterparts. Meanwhile, in these systems, the ordered parallel line arrays containing planar and rigid π-π structures would promote electron delocalization and improve conjugation degree. Recently, Luh et al. [32,33,34,35,36,37] reported various double-stranded polynorbornenes via ring-opening metathesis polymerization (ROMP) technique, and multifarious aromatic moieties were designed as the linkers between two main-chains. In addition, we recently reported perylene diimides containing single- and double-stranded PAs by CP method, which generated ordered two-dimensional architecture, greatly improving the overall photophysical properties, such as broadened light absorption, lower LUMO energy level [10]. In this account, more relevant systems should be envisaged to expand the scope of monomers, and AB moieties are then selected as the pendants or linkers for the corresponding single- and double-stranded PAs.

Herein, we designed and synthesized AB-linked bis(1,6-heptadiyne) and 1,6-heptadiyne derivatives as monomers for CP to create double-stranded **P1** and the corresponding single-stranded **P2** (Scheme 1). The behavior of the *trans-cis* photoisomerization process and the further influence on the photophysical properties of the PA backbone were also investigated.

## 2. Results and Discussion

### 2.1. Synthesis and Characterization of AB-linked 1,6-Heptadiyne Monomers

The AB-linked 1,6-heptadiyne monomers **M1** and **M2** with different chain lengths were synthesized by classical esterification reaction of 4,4′-dihydroxyazobenzene with compound **1** and Williamson reaction of compound **2** with corresponding aromatic 4-hydroxyazobenzene, as shown in Scheme 1. Monomers **M1** and **M2** were designed not just because these compounds could be easily synthesized but also because the introduction of AB moiety to ladder architecture together could create new ladder-like conjugated PAs with multi-planner linkers. We believe it is rational and achievable to prepare these ladder-like polymers via CP of AB-linked bis(1,6-heptadiyne) using **Ru-III** as the catalyst. The structures of **M1** and **M2** were characterized by ^1^H NMR and HR-ESIMS spectra. We chose CDCl_3_ as a solvent for ^1^H NMR measurement. As shown in Figure 1a of **M1**, the two multiplet peaks at 7.95–7.84 ppm and 7.54–7.47 ppm were coming from the aromatic region of azobenzene, and the chemical shifts of CH_2_ protons near to ester group appeared at 4.25–4.17 ppm. The terminal alkyne protons appeared at 2.01–1.97 ppm. The residual protons appeared at 2.94–2.83 ppm, 2.81–2.71 ppm, and 2.24–2.13 ppm for **M1**. For monomer **M2** (Appendix A), there existed four multiplet peaks at 7.95–7.84, 7.54–7.47, 7.46–7.41, and 7.05–6.98 ppm for AB moiety and triplet peaks at 2.01–1.97 ppm for terminal alkyne protons. Besides, the ratio of the integrated areas of the peaks at 2.01–1.97 ppm and 7.95–7.84 ppm was 1:1 (**M1**) and 1:2 (**M2**), respectively, which was consistent with the proton number of the expected compounds. Meanwhile, the ESI–MS data of M+Na were 645.2243 for **M1** and 397.1593 for **M2**, respectively, which matched well with the theoretical calculating values. All these characteristic data confirmed the successful synthesis of AB-linked 1,6-heptadiyne monomers.

### 2.2. Synthesis and Characterization of AB-Linked Double- and Single-Stranded ***P1*** and ***P2***

With monomers **M1** and **M2** in hand, the next stage was to synthesize the relative polymers, and the results are collected in Table 1. As an initial attempt, MCP of **M1** was carried out in a weakly coordinating solvent Tetrahydrofuran (THF) at 0 °C with a molar feed ratio of **M1** to initiator **Ru-III** ((M)/(I)) of 10 and 40 (Runs 1–4). When **Ru-III** was added to the solution of **M1**, the orange color gradually changed to dark-red after 30 min, and finally, a shiny purple-black solid polymer, the double-stranded **P1**, was obtained by precipitating the mixture from excess acetone in high yield. However, with increasing monomer loading, the higher(M)/(I) ratio of 50 (Runs 5) was attempted with an expectation to obtain higher molecular weight polymers, whereas the maximum degree of polymerization (DP) was only up to 42, even prolonged reaction time to 24 h, and the obtained dried powder solid **P1** was no longer soluble in THF, CHCl_3_, CH_2_Cl_2_, and o-dichlorobenzene, which might be attributed to the stronger π-π stacking behavior between AB linkers in molecular chains. Because of the poor re-solubility of **P1** in higher molecular weight in these common organic solvents, only **P1** in lower molecular weight was characterized by NMR and Gel permeation chromatography (GPC) techniques.

Similarly, for the synthesis of single-stranded **P2**, when the molar ratio of (M)/(I) was from 25 to 50 (Runs 6–7), the resultant polymer had the highest DP of 48. Increased (M)/(I) ratios from 75 to 100 were then used (Runs 8–9), and the corresponding polymers had slightly higher DPs of 60 and 78 in decreased yields of 80% and 65%, respectively. Contrarily, the obtained dried polymer **P2** could be easily re-dissolved in THF, CHCl_3_, CH_2_Cl_2_, and o-dichlorobenzene. The structures of representative polymers were characterized by ^1^H NMR spectroscopy. As compared with **M1**, the proton signals at 2.01–1.97 ppm (Figure 1a) assigned to the terminal alkenes disappeared after CP, and a symmetric broad peak observed at 6.52 ppm for the polyene protons (H_a_, Figure 1b) on the conjugated double bonds of double-stranded **P1**, implying that the double bonds of **P1** have the same configuration [12,38]. Besides, the broad peaks around 2.58–2.02 ppm were assigned to the saturated protons (H_b+c_) of the repeat five-member rings. Similar results were also found for single-stranded polymer **P2**.

### 2.3. Morphology of Ordered Ladder Architecture

The TEM images of **P1** (Figure 2a,b and Appendix A) showed a well-aligned motif on the copper mesh. Obviously, the ladder conformation was fully verified. It can be seen that lots of black stripes were arranged in parallel with each other. The apparent width of each stripe was about 0.15 nm, which would match nicely to the width of the AB unit in poly(1,6-heptadiyne)-based ladderphanes.

This result illustrated that aromatic AB chromophores would align face-on orientation with respect to the substrate surface. Attempts to observe molecular images for ladderphane nanostructure were unsuccessful because of the negligible contrast between adjacent ladders. The present large scale of ladder architecture indicated that the assembly of the polymeric molecules might also exhibit π−π attractions between end groups (vinyl and styryl) along the longitudinal axis of the polymers and van der Waals interactions between the neighboring polymeric backbones [10], as shown in Figure 2c,d. Interestingly single-stranded **P2** (Figure 3) showed micelles morphology with size mostly in the range of 50–80 nm, which is similar to that of azobenzene-based micelles [39]. In the spheres formed by self-assembly, the size observed in Dynamic Light Scattering (DLS) increased and was in the range of 60–120 nm, which might be attributed to the solvation of polymer in solution.

### 2.4. Photoisomerization Behaviors of ***P1*** and ***P2***

UV absorption spectroscopy was performed to provide optical properties of the fully conjugated polymer structure and examine the photoisomerization of AB moiety in the THF solution. In Figure 4a, **M1** exhibited strong UV-vis absorption bands at approximately 329 nm, which were attributed to the π–π* transition in AB moiety’s *trans*-isomer. Meanwhile, weak absorption bands at approximately 437 nm were engendered by an n–π* transition in the AB *cis*-isomer [9,10]. While, the π–π* and n–π* transition of **M2** showed a nearly 15 nm red-shift (Figure 4b), which was possibly induced by the different groups connected to AB moiety. The monomers **M1** and **M2** achieved a *trans*-*cis* photostationary state at the beginning of irradiation within 60 s and 30 s, respectively. While the limited conversion space and the intense restraint between two 1,6-heptadiynes eventually led to incomplete and sluggish *trans*-*cis*-*trans* photoisomerization of **M1** (Figure 4a and Appendix A). It was worth noting that **M2** displayed distinctive photoresponse compared with other AB-containing compounds [1,2,3,4,5,6,7,8], of which *trans*-*cis*-*trans* photoisomerization could complete rapidly without the space restriction and chain constraint, as illustrated (Figure 4b and Appendix A). Simultaneously, it exhibited a 30 nm blue shift (345 nm to 315 nm) under UV light and a reversible 30 nm redshift (315 nm to 345 nm) under visible light.

However, after the MCP reaction, the newly formed conjugated PAs backbones on **P1** and **P2** evolved at higher wavelengths 400–650 nm and overlapped with AB *cis*-isomer, which consequently caused the difficulty to detect *trans*-*cis*-*trans* isomerization. For double-stranded **P1**, as time proceeded, the original rose-carmine solution gradually faded until colorless within 20 min (Appendix A). Correspondingly, the λ_max_ gradually blue-shifted from 545 nm to 430 nm, and the absorption overall intensity decreased, indicating the shortened conjugated length [10]. Interestingly, the UV−Vis spectra of **P2** divided into two periods: Firstly, after UV light irradiation for 5–30 s, significant 50 nm red shift and enhanced overall intensity at 554 nm were obviously observed (Figure 4d). These results suggested the straightening of the polyene backbone, which finally led to an increased conjugated length. Secondly, by extending irradiation time (20 s–40 min), **P2** displayed a similar photoresponse with that of **P1**.

### 2.5. Thermal Stabilities of ***P1*** and ***P2***

The thermal stability of these single- and double-stranded poly(1,6-heptadiyne)s was investigated by means of thermal gravimetric analysis (TGA). Generally, double-stranded polymers have greater resistance to irradiation, as well as thermal and chemical degradation, in comparison to their single-stranded counterparts [12]. As shown in Figure 5, double-stranded **P1** degraded into two steps under a nitrogen atmosphere. The initial decomposition temperature was 304 °C, the second degradation step started at 456 °C, and 10 wt.% loss temperature (T_d_) reached to 417 °C even when the polymer was heated to 800 °C, and 55% of the weight was still retained, indicating the excellent thermal stability of double-stranded **P1** [3a]. While single-stranded **P2** began to decompose at 207 °C, T_d_ was only 326 °C, and it retained 25% of its original weight at 800 °C, that is, the half weight of **P1** residue, which is also important for the practical application of polymers, especially when they are used under high temperature.

## 3. Materials and Methods

### 3.1. Materials

4-hydroxyazobenzene and 4,4′-dihydroxyazobenzene were purchased from Shanghai DiBo Chemical Technology Co., LTD (Shanghai, China). 4-Hydroxymethyl-1,6-heptadiyne and 2-(2-propinyl)-4-pentynoic acid were prepared by the same method from the previous literature [13]. Dichloro[1,3-bis(2,4,6-trimethylphenyl)-2-imidazolidinylidene](benzylidene)bis(3-bromopyridine)ruthenium(II) (Grubbs third-generation catalyst, **Ru-III**), succinic anhydride, and 3-Bromo-1-propanol were obtained from Aldrich. Ethyl vinyl ether (stabilized with 0.1% *N*,*N*-diethylaniline) was purchased from Acros. 4-dimethylaminopyridine (DMAP) and 1-(3-dimethylaminoprop-yl)-3-ethylcarbodiimidehydrochloride (EDCI·HCl) were purchased from Energy Chemical (Shanghai, China). All reactions were carried out under a dry nitrogen atmosphere using standard Schlenk-line techniques. Solvents were distilled over drying agents under nitrogen prior to use: dichloromethane (CH_2_Cl_2_) from calcium hydride, THF from sodium wire. All solvents and other chemical reagents were purchased from Shanghai Chemical Reagent Co., Ltd. (Shanghai, China) and used as received without any further treatment. ^1^H (500 MHz) spectra were recorded using tetramethylsilane as an internal standard on a Bruker DPX spectrometer. The HR-ESIMS was measured by a Bruker QTOF micromass spectrometer. UV-vis absorption spectra were measured on a UV-1800 spectrometer. GPC was used to calculate relative molecular weight and molecular weight distribution equipped with a Waters 1515 Isocratic HPLC pump, a Waters 2414 refractive index detector, and a set of Waters Styragel columns (7.8 × 300 mm, 5 mm bead size; 10^3^, 10^4^, and 10^5^ Å pore size). GPC measurements were carried out using THF as the eluent with a flow rate of 1.0 mL/min. The system was calibrated with a polystyrene standard. UV–Vis absorption spectra were measured on an Agilent Cary 60 spectrometer (Agilent Technologies Inc., California, CA, USA). The UV irradiation was carried out with an 8 W × 4 UV lamp with the wavelength at 365 nm. TGA was performed using an SDTA851e/SF/1100 TGA Instrument under nitrogen flow at a heating rate of 10 °C/min from 50 to 800 °C. Samples for transmission electron microscopy (TEM) were prepared by depositing a drop of the solution (1 mg/mL) on a carbon-coated Cu grid, and TEM images were recorded on the JEOL2100F microscope operating at 120 kV (Japan Electronics Co., Ltd., Tokyo, Japan).

### 3.2. Synthesis of Compound ***1***

To a solution of 4-hydroxymethyl-1,6-heptadiyne (6.1 g, 50 mmol) in 60 mL of CH_2_Cl_2_, succinic anhydride (6 g, 60 mmol) and DMAP (1.22 g, 10 mmol) were added and stirred under an N_2_ atmosphere for 24 h at room temperature. The CH_2_Cl_2_ phase was washed with 1 M HCl and saturated saltwater until the water phase reached neutrality. After that, the CH_2_Cl_2_ solvent was evaporated, and the crude product was purified by column chromatography on silica gel using petroleum ether/ethyl acetate (1:5) as eluent. The compound **1** was obtained as a yellow sticky liquid (10.2 g, 89%); ^1^H NMR (500 MHz, CDCl_3_, ppm): δ 8.87 (s, 1H, -COOH), 4.21 (d, 2H, -CH_2_O°C-), 2.72 (m, 2H, -CH_2_COOH), 2.65 (m, 2H, -CH_2_CH_2_COO-), 2.39 (m, 4H, -CHCH_2_C≡CH), 2.15 (m, 1H, -CH_2_CHCH_2_-), 2.03 (t, 2H, C≡CH).

### 3.3. Synthesis of AB-Linked Bis(1,6-heptadiyne) Monomer ***M1***

Compound **1** (1.66 g, 8 mmol) was firstly dissolved in 50 mL of anhydrous CH_2_Cl_2_. To this solution, 4,4′-dihydroxyazobenzene (0.80 g, 3.5 mmol), EDCI·HCl (2.20 g, 12 mmol), and DMAP (0.09 g, 0.8 mmol) were added under an N_2_ atmosphere in an ice-water bath for 1 h, and then the reaction progress proceeded at room temperature and was monitored by TLC. After 4 days, the mixture was evaporated to remove CH_2_Cl_2_ and then poured to 400 mL of water to precipitate dark red solid. The solid was purified by column chromatography on silica gel using CH_2_Cl_2_ as eluent. The product **M1** was obtained as an orange needle-like powder (1.49 g, 70%, Rf value is 0.35); ^1^H NMR (500 MHz, CDCl_3_, ppm): δ 7.95–7.84 (m, 4H, CHCN=N), 7.54–7.47 (m, 4H, °CCH), 4.25–4.17 (m, 4H, CO°CH_2_), 2.94–2.83 (t, 4H, CH_2_°C°CH_2_CH_2_), 2.81–2.71 (t, 4H, CH_2_°C°CH_2_CH_2_), 2.47–2.31 (t, 8H, CHCCH_2_), 2.24–2.13 (m, 2H, CHCH_2_O), 2.01–1.97 (t, 4H, CHCCH_2_); ESI-MS: Calcd. For C_36_H_34_N_2_O_8_ Na [M+Na]^+^: 645.2315, Found: 645.2243.

### 3.4. Synthesis of Compound ***2***

To a solution of 2-(2-propinyl)-4-pentynoic acid (4.08 g, 30 mmol) and 3-Bromo-1-propanol (4.17 g, 30 mmol) in 50 mL anhydrous CH_2_Cl_2_, EDCI·HCl (4.35 g, 36 mmol) and DMAP (0.47 g, 0.4 mmol) were added under stirring at ice-water bath for 2 h, and then the reaction progress proceeded at room temperature and was monitored by TLC. After 1 day, the mixture was washed with water, dried with MgSO_4_, and evaporated to remove CH_2_Cl_2_. The solid was purified by column chromatography on silica gel using CH_2_Cl_2_/petroleum ether (1:1) as eluent. The product **2** was obtained as sticky liquid and used without any further characterization (6.9 g, 90%) except for ESI-MS. ESI-MS: Calcd. For C_11_H_13_O_2_BrNa [M + Na]^+^: 279.0099, Found: 279.0107.

### 3.5. Synthesis of AB-Linked 1,6-Heptadiyne Monomer ***M2***

4-hydroxyazobenzene (1.58 g, 8 mmol), potassium carbonate (6.21 g, 45 mmol), and 40 mL of *N*,*N*-Dimethylformamide (DMF) were charged into a 250 mL Schlenk flask. The reaction mixture was heated at 80 °C for 8 h under nitrogen atmosphere, allowing for the formation of potassium salt. A solution of compound **2** (3.07 g, 12 mmol) in 10 mL of DMF was then added dropwise to the above mixture. After 24 h of stirring at 60 °C, the reaction mixture was poured into excess water, and the crude product was precipitated out and further purified by chromatographic purification (silica gel, CH_2_Cl_2_/petroleum ether 1:2) to give yellow solid (2.42 g, 82%, Rf value is 0.54). ^1^H NMR (500 MHz, CDCl_3_, ppm): δ 7.95–7.84 (m, 4H, N=NCCHCHCH), 7.54–7.47 (m, 2H, N=NCCHCHCH), 7.46–7.41 (m, 1H, N=NCCHCHCH), 7.05–6.98 (m, 2H, °CCH), 4.42–4.35 (t, 2H, CH_2_OAr), 4.20–4.12 (t, 2H, CO°CH_2_), 2.85–2.74 (m, 1H, °C°CH), 2.68–2.62 (m, 4H, CH_2_CHCOO), 2.24–2.15 (m, 2H, °CH_2_CH_2_CH_2_O), 2.01–1.97 (t, 2H, CHCCH_2_); ESI-MS: Calcd. For C_23_H_22_N_2_O_3_ Na [M + Na]^+^: 397.1630, Found: 397.1593.

### 3.6. General Procedure for Polymerization

Typically, polymerization was carried out in a Schlenk tube under nitrogen atmosphere at 0 °C in THF for a preset time. After confirming the monomer conversion by TLC, a small amount of ethyl vinyl ether was added to the mixture and stirred for 0.5–3 h; the reaction mixture was concentrated and poured into an excess of acetone. The dark red polymer was washed successively with acetone and dried in a vacuum oven at 40 °C to a constant weight.

#### 3.6.1. Synthesis of AB-Linked Double-Stranded **P1**

A typical polymerization procedure using **Ru-III** initiator was as follows: A 100 mL Schlenk tube was charged with monomer **1** (124 mg, 0.12 mmol) dissolved in 40 mL of THF. In another 10 mL Schlenk tube, **Ru-III** (11 mg, 0.012 mmol) was dissolved in THF (1 mL). After degassing with three freeze-vacuum-thaw cycles, the catalyst solution of **Ru-III** was then injected into the monomer solution via a syringe under vigorous stirring at 0 °C for 0.5 h. ^1^H NMR (500 MHz, CDCl_3_, ppm): δ 7.87–7.71 (m, N=NCCHCHCH), 7.49–7.25 (m, N=NCCHCH), 6.52 (s, conjugated CH=CH on the backbone), 4.37–4.12 (m, CO°CH_2_), 3.23–2.67 (m, °C°CH_2_CH_2_COO), 2.58–2.02 (m, CH_2_CHCH_2_).

#### 3.6.2. Synthesis of AB-Linked Single-Stranded **P2**

The synthesis of AB-linked single-stranded **P2** was similar to that of **P1.** Monomer **M2** (112 mg, 0.3 mmol) and **Ru-III** (13 mg, 15 μmol) were stirred in 1 mL of THF at 0 °C for 3 h. ^1^H NMR (500 MHz, CDCl_3_, ppm): δ 7.93–7.81 (m, N=NCCHCHCH), 7.49–7.34 (m, N=NCCHCHCH), 6.9–6.84 (m, °CCHCH), 6.52 (s, conjugated CH=CH on the backbone), 4.4–4.02 (m, °CH_2_CH_2_CH_2_O), 3.2–2.71 (m, CH_2_CHCH_2_ on the five-member ring), 2.2–1.92 (m, °CH_2_CH_2_CH_2_O).

## 4. Conclusions

In summary, we demonstrated double-stranded poly(1,6-heptadiyne) and single-stranded poly(1,6-heptadiyne) using AB chromophore via CP reaction. This methodology enabled us to obtain the poly(1,6-heptadiyne)s with controlled nanostructure, good solubility, wide absorption range, and good thermal stability. The double-stranded **P1** self-associated through π-π interaction to produce controlled ladderphane nanostructures, and the single-stranded **P2** formed micelles. The structures and properties of these polymers were characterized fully by spectroscopic means. The **M1** and **M2** showed characteristic *trans*-*cis*-*trans* photoisomerization upon irradiation with 365 nm UV light and visible light. After CP reaction, the newly formed conjugated PAs backbones on **P1** and **P2** broadened UV-Vis absorption range to 650 nm; because of the photo-oxidation and loss of the conjugation in the backbone, the *trans*-*cis*-*trans* photoisomerization could hardly be observed. Further improvement of film-forming property and investigation on the photovoltaic performance of polymers are still underway in our laboratory.

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
