# Peer review of "Metathesis Cyclopolymerization Triggered Self-Assembly of Azobenzene-Containing Nanostructure"

_molecules, 2020, doi:10.3390/molecules25173767_

Round 1
Reviewer 1 Report
Song and coauthors report on the synthesis and photochemical properties of two monomeric units containing azobenzene moieties along with the corresponding polymeric derivatives obtained after metathesis cyclopolymerization.
The study is solid however it needs a revision of the following points before acceptance in Molecules
Major points:
- The authors report the photochemical isomerization of azobenzene with 360 nm wavelength. The efficiency of the isomerization of azobenzene is extremely dependent on the wavelength used, due to its non-Kasha behavior. The authors should try to isomerize M1 and M2 with different wavelengths (like 310nm for M1) to obtain better photoconversion.
- The distribution of cis and trans isomers at the photostationary state should be measured for M1 and M2.
- The thermal lifetimes for M1 and M2 should be measured, too.
- The authors should explicitly state that the photoisomerization properties of the azo chromophore are lost in the polymer.
Considering the rigidity of P1 and P2 and the UV-vis it is more probable that irradiation just ends up in photooxidation and loss of the conjugation in the backbone.
It is possible that the trans-cis isomerization is not operative. In any case, the UV-vis reported here do not provide a conclusive evidence for a clean photoisomerization. Please change the sentences referring to the isomerization properties in the properties in the body of the text and in the conclusions. - No Supporting Information file is furnished in the submission, even though it is explicitly mentioned in the draft. Please provide the file
Minor points:
- Abstract line21: remove While
- The linear representation of azobenzene is intrinsically wrong.
I would suggest changing the abbreviation of azobenzene from Azo as used in the text to AB, to avoid confusion with the azo chromophore (-N=N-) - The description of the abbreviation of azobenzene should be present also in the introduction and not only in the abstract
- Line 30 The should be the. Wherein sounds a bit odd in the first sentence, which is also quite long and not extremely readable. Consider rewriting
- Line 32, remove While
- Line 41, more general reviews on azobenzenes should be added:10.1039/C1CS15179G 10.1038/s41570-019-0074-6
- Line 61, Account should be account
- Line 83, Specify that the signals are coming from the aromatic region of azobenzene, instead of writing Azo-H.
- Line 84, eater should be ester
- Line 85, put a space between 2.71 and ppm
- Figure 1, Please furnish the original NMRs with integration in the SI. What is the peak at 0? grease? If so the yields should be double-checked
- Line 141, 0.32 nm: I doubt that an azobenzene could be as long as 0.32 nm. Azobenzene is normally one third of that width in its trans form. Please explain
- Figure 2c and d, put a box in the figure to highlight that the azobenzene chromophore is represented by a green sphere
- Figure 2b, consider changing the colors used. The readability with the current colors is extremely low
- 10.1021/acs.langmuir.6b04455 should be added as a reference for azobenzene based micelles
- Line 163, moiety' lacks an s
Author Response
Thanks for the reviewer’s comments, we have added some contents in the revised manuscript according to your suggestions.
-
At first, we also want to test the trans-cis transformation behavior using monochromatic light source, unfortunately, due to the limited experimental conditions, we can just use a common ultraviolet light which can provide 365 nm light. So, it is very sorry that the transformation behavior stimulated under 310 nm may not be added. However, we do appreciate your great suggestion and also hope to possess improved testing tools in the near future.
-
The distribution of cis and trans isomers at the photostationary state was added in Figure S2 a,b.
-
The thermal decomposition temperature of AIBN located at nearly 65℃, the M1 and M2 monomers possibly completely decomposed below 300℃. Because we never test the thermal stability of small molecules before, and mainly because of the 2019-nCoV, we are absolutely not allowed into school to do experiments during the summer vacation. Thus, it is very difficult to provide the weight loss of M1 and M2.
-
The sentences referring to the isomerization properties in the properties in the body of the text and in the conclusions were changed.
-
Abstract line21: While was removed.
-
The abbreviation of azobenzene from Azo as used in the text to AB. And the description of the abbreviation of azobenzene was present also in the introduction.
-
Line 30 the first sentence was rewriten.
-
Line 32, While was removed.
9. Two reviews on azobenzenes were added:
Bandara, H. M. D.; Burdette, Shawn. C. Photoisomerization in different classes of azobenzene. Chem. Soc. Rev., 2012, 41, 1809-1825.
Crespi, S.; Simeth, N. A.; König, B. Heteroaryl azo dyes as molecular photoswitches. Nat. Rev.Chem. 2019, 3, 133-146.
10. Line 61, Account was changed to be account.
11. Line 83, the sentence “the two multiplet peaks at 7.95-7.84 ppm and 7.54-7.47 ppm were assigned to Azo-H”was changed to be “the signals are coming from the aromatic region of azobenzene”.
12. Line 84, eater was changed to be ester.
13. Line 85, a space between 2.71 and ppm was added.
14.The original NMRs with integration was supplied here, take the M2 as example,the peak at 0 ppm was single peak, and there has two peaks beside 0 ppm in our previous NMRs spectrum, which may be caused by the different CDCl3. We just sent the solid samples to other university for test, as for the CDCl3, we don’t know if the batches of deuterated chloroform are the same and also the details of test process.
15. .0.32 nm was the width of bright stripe and neighboring two dark zone, not just the width of Azobenzene, so 0.32 nm was not correct. The stripe was nearly 0.15 nm.
-
Figure 2c and d, a box in the figure to highlight that the azobenzene chromophore is represented by a green sphere was added.
-
The colors was changed to yellow as white text will reduce the readability.
-
A referencre was added:
Filipová, L.; Kohagen, M.; Štacko, P.; Muchová, E.; Slavíček, P.; Klán, Petr. Photoswitching of Azobenzene-Based Reverse Micelles above and at Subzero Temperatures As Studied by NMR and Molecular Dynamics Simulations. Langmuir 2017, 33, 2306-2317.
-
moiety' lacks an s, and the s was added.
Reviewer 2 Report
This article describes the preparation of new conjugated polymers incorporating an azo-benzene group. These polymers present interesting self-assembly properties: the single-stranded one forms micelles whereas the double-stranded one self-associates through pi-pi interactions to produce controlled ladderphane nanostructures. The thermal stability of new materials and photoisomerization studies are also described.
The results are well-presented and are of great interest for future applications in material sciences. However, some corrections/additions are needed to improve the article:
- Scheme 1: the reagents for the reaction from bromide 2 to monomer M2 are not correct (should be K2CO3/DMF instead of EDCI/DMAP)
- lines 80 and 168: "1,6-heptadiyne"
- line 84: "ester group"
- line 104: "powder solid"
- line 153: "attributed to the"
- line 169: figure 4a instead of 3a
- line 190: "these results"
- lines 213 and 261: "pentynoic"
- M1 and M2 are molecules never described in the literature before. It seems to me that characterizing them only with HRMS and NMR-1H is not sufficient (page 8). Complementary analyses should be given in order for others to be able to easily reproduce the results. At the minimum, NMR-13C should be added. IR and Rf (as the two monomers were purified by chromatography) could also be given.
- page 8, compound 2: NMR-1H should be added to fully characterize the compound 2. The formula used for the HRMS analysis is not correct (should be C11H13O2BrNa).
- page 8, compound M2: the yield of the reaction is not given and should be added.
Author Response
Thanks for the reviewer’s comments, we have added some contents in the revised manuscript according to your suggestions.
- Scheme 1, the reagents for the reaction from bromide 2 to monomer M2 are changed to be K2CO3/DMF.
- 1,6-haptadiyne was changed to be 1,6-heptadiyne.
- easter was corrected to be ester.
- power was corrected to be powder.
5.pentinoic was corrected to be pentynoic.
- M1 and M2 was purified by chromatography, mainly because of the chemical plant explosion in my city of Yancheng and the 2019-nCoV, we are not permitted to do any experiment during summer vacation. It is very sorry that there is only one nuclear magnetic resonance hydrogen spectrometer in another yancheng teachers university, and test was stopped during summer vacation and the deadline is coming, so we could not supply 13 C NMR spectra of M1 and M2, and 1 H NMR of compound 2 promptly before the deadline. Thanks for your suggestion, I will supply sufficient evidences to verify the structure in the future. So we just added the Rf values. Please forgive us for the inconvenience.
- The formula of compound 2 was corrected to be C11H13O2BrNa.
- The yield of M2 was added.
Round 2
Reviewer 1 Report
The authors modified the manuscript to the best of their possibilities and provided an improved version of their work. The manuscript can be accepted as is.
Reviewer 2 Report
Although some of my concerns could not be addressed due to the situation, I think that the improvements made by the authors are sufficient to enable the publication of this article in its present form.
This manuscript is a resubmission of an earlier submission. The following is a list of the peer review reports and author responses from that submission.